# Does the Presence of Matted Nodes in Colon Adenocarcinoma Influence 5-Year Overall Survival?

**DOI:** 10.3390/medicina60081194

**Published:** 2024-07-24

**Authors:** Karla I. Rodríguez-López, Mariana Salazar-Castillo, Leonardo S. Lino-Silva, Ángeles Galán-Ramírez, Luisa F. Rivera-Moncada, Emiliano A. López-Jiménez, César Zepeda-Najar

**Affiliations:** 1Surgical Pathology, National Cancer Institute (Mexico), Tlalpan 14080, Mexico City, Mexico; karlaisarolo07012002@gmail.com (K.I.R.-L.); mariana.salazar.cas@gmail.com (M.S.-C.); angelesgalan.rmz@gmail.com (Á.G.-R.); luisafernandarivera08@gmail.com (L.F.R.-M.); 2AFINES Program, Medicine Faculty, National Autonomus Universiti of Mexico (UNAM), Coyoacán 04510, Mexico City, Mexico; emilianolopezjimenez@gmail.com; 3Surgical Oncology, Ángeles Tijuana Hospital, Tijuana 22010, Baja California Norte, Mexico; cz.surgicaloncology@gmail.com

**Keywords:** matted nodes, lymph node metastasis, colon cancer, rectal cancer, colorectal cancer, overall survival

## Abstract

*Background and Objectives*: Colon cancer (CC) is prevalent globally, constituting 11.9% of cases in Mexico. Lymph node metastases are established prognostic indicators, with extracapsular lymph node extension (ENE) playing a crucial role in modifying prognosis. While ENE is associated with adverse factors, certain aspects, like matted nodes (lymph node conglomerates), are underexplored. Matted nodes, clusters of lymph nodes infiltrated by cancer cells, are recognized as an independent prognostic factor in other cancers. This study investigates the prognostic implications of matted nodes in CC. *Materials and Methods*: From a retrospective analysis of 502 CC consecutive cases treated with colectomy (2005–2018), we identified 255 (50.8%) cases with lymph node metastasis (our study group), which were categorized into two groups: (1) lymph node metastasis alone (n = 208), and (2) lymph node metastasis with matted nodes (n = 47). A comparative survival analysis was performed. *Results*: Of the 255 patients, 38% had lymph node metastasis. Patients with matted nodes (18.4%) showed an association with higher pN stage and lymphovascular invasion. The 5-year survival rate for patients with matted nodes was 47.7%, compared to 60% without (*p* = 0.096); however, this association demonstrated only a statistical tendency. Multivariate analysis identified clinical stage and adjuvant chemotherapy use as independent factors contributing to survival. *Conclusions*: This study underscores matted nodes as potential prognostic indicators in CC, emphasizing their association with higher pN stage and reduced survival. Although the patients with matted nodes showed lower survival, this figure did not search statistical significance, but a tendency was detected, which necessitates precise further research, which is essential for validating these findings and integrating matted nodes into the broader context of colorectal cancer management.

## 1. Introduction

Colon cancer (CC) ranks among the most prevalent malignancies worldwide. In Mexico, recent data from Globocan reveals that CC accounts for 11.9% of cases, affecting 6.1% of men and women, respectively. [1] The presence of lymph node metastases in CC is a well-established prognostic indicator. Beyond the mere existence of lymph node metastases, specific characteristics within these metastases can further influence prognosis. Notably, extracapsular lymph node extension (ENE), as previously elucidated [2], plays a pivotal role in modifying prognosis. Various studies underscore the significance of ENE in metastatic lymph nodes across different malignancies. Drawing from the limited evidence available, it is evident that extracapsular LNI is a prevalent occurrence in patients with gastrointestinal malignancies. Furthermore, ENE has been linked with younger age, advanced tumor stage, lymphovascular invasion (LVI), and perineural invasion (PNI) [3].

However, certain aspects of lymph node metastases in CC, such as lymph node conglomerates (matted nodes), remain unexplored. Matted nodes denote the presence of clusters of lymph nodes infiltrated by cancer cells, causing them to merge with surrounding tissue and among neighboring lymph nodes. The presence of metastatic lymph nodes and their quantity are pivotal indicators for staging and prognosis, furnishing vital insights into the cancer’s aggressiveness and potential treatment strategies. Nevertheless, it remains uncertain whether the presence of matted nodes exerts a detrimental impact on patients who already harbor lymph node metastases. In other types of cancer, matted nodes (nodal conglomerates) have been identified as an independent prognostic factor associated with adverse outcomes. For instance, a systematic review indicates that patients with matted nodes face up to a 1.6-fold higher risk of mortality compared to those without [4]. Matted nodes are considered an adverse prognostic factor in various cancers for several reasons; the presence of matted nodes often signifies a more advanced stage of the disease. This suggests that the cancer has spread extensively to the lymph nodes, complicating treatment and reducing the likelihood of remission [5]; matted nodes indicate a higher tumor burden, this makes it more challenging to eradicate the cancer completely through treatments like surgery, radiation, or chemotherapy [6]. Also, cancers with matted nodes are often more resistant to conventional treatments. The conglomeration of lymph nodes can be a sign of tumor aggressiveness and that the cancer cells possess biological characteristics that enable them to survive therapies [7], and matted nodes can facilitate the spread of cancer to other parts of the body. The fusion of lymph nodes indicates that cancer cells have a greater ability to invade and colonize other tissues [8].

Given the dearth of research on the prognostic implications of matted nodes in CRC, our objective was to ascertain whether their presence independently correlates with reduced survival in patients diagnosed with this form of cancer.

## 2. Materials and Methods

Ethical considerations. This study was approved by our institutional research and ethics board (approval number 2023/085), which is available upon reasonable request. This study was conducted in accordance with the Declaration of Helsinki ethical principles.

Population. This is a retrolective observational study identifying all consecutive cases of CRC treated with surgery at our institution spanning from 2005 to 2018, culminating in a cohort of 502 patients. A stringent selection process was applied, selecting only the cases presenting with lymph node metastasis; cases with evidence of distant metastasis at initial presentation were excluded, resulting in a refined subset of 255 cases, which correspond to our study group. These cases were further categorized based on their lymph node metastasis status into two groups: (1) those with lymph node metastasis alone (n = 208), and (2) those with lymph node metastasis accompanied by matted nodes (n = 47). Matted nodes were meticulously defined in alignment with the National Cancer Institute’s criteria, characterized as a cluster of fused lymph nodes validated through histopathological examination (https://www.cancer.gov/publications/dictionaries/cancer-terms/def/matted-lymph-nodes, accessed on 1 May 2024). This confirmation was established by the presence of two or more contiguous nodes sharing and adhered by neoplastic cells in their parenchyma and capsules (see Figure 1).

Clinical features. Relevant clinical and pathological data, alongside established prognostic factors, were systematically collected from the clinical and pathologic records of our institution. The comprehensive dataset included information on sex, age, tumor location, histological type, pathologic T stage, pathologic N stage, surgical margins, distant metastases (identified during follow-up), clinical stage, utilization of adjuvant treatment, follow-up duration in months, and patient outcomes (alive or deceased). Following data collection, a meticulous histopathological review was conducted to authenticate the pathological variables and categorize the cases accordingly.

Statistical analysis. Upon data compilation, a thorough descriptive analysis of the population was executed, summarizing numerical variables with median and interquartile range and categorical variables with percentages. Subsequently, a bivariate analysis was undertaken, comparing clinicopathological characteristics among the three groups. Numerical variables were scrutinized using the Student’s *t*-test, while categorical variables underwent analysis via chi-square testing. A 5-year survival analysis, employing the Kaplan–Meier method, was conducted, comparing clinical and pathological characteristics recognized in the literature as being associated with prognosis, including the study groups. Generated survival curves were subjected to comparison using the log-rank test. Further analyses, stratified by clinical stage, and a multivariate analysis using Cox regression were performed. A significance level of *p* < 0.05 was employed for all analyses. The statistical software SPSS 29.0 (IBM, Armonk, New York, NY, USA, 2022) facilitated these comprehensive analyses.

## 3. Results

From a dataset encompassing 502 consecutive patients who underwent colectomy at our institution between 2010 and 2015, 255 cases (38%) with lymph node metastasis were discerned. These patients had a mean age of 57.57 years, spanning from 21 to 88 years, with 54.1% being women and 45.9% men. Predominantly, a significant proportion of patients were in clinical stage III (64.7%), while the remainder were in stage IV. Notably, 166 patients (65%) were still alive at an average follow-up duration of 43 months.

Concerning clinicopathologic features, the distribution of cases based on tumor location revealed a dominance in the right colon (66.3%), followed by the sigmoid (18%), descending colon (11.8%), and transverse colon (3.9%). The median tumor size was 60 mm, with 90% of cases being of the conventional type. Pathologically, 128 cases (50.2%) were classified as stage T3, 89 (34.9%) as stage pT4a, 28 (11%) as stage pT4b, and 10 (3.9%) as stage pT2. The most prevalent histologic grade was G2 in 123 cases (48.2%), followed by G3 in 111 cases (43.5%), and G1 in 21 cases (8.2%). Lymphovascular invasion was observed in 62.4% of cases, while 39.6% exhibited PNI. The median number of lymph nodes resected was 23 (ranging from 12 to 85), and the median number of lymph nodes with metastasis was 4 (ranging from 1 to 51).

Of the 255 patients, 47 (18.4%) presented with matted nodes. A summary of the clinical and pathological characteristics of these patients, categorized by the presence of matted nodes, is presented in Table 1. The table indicates an association between matted nodes and a higher pN stage, along with an association with LVI.

Table 2 details the factors linked to survival among patients with lymph node metastasis. Factors such as pN stage, clinical stage, LVI, PNI, and the utilization of adjuvant chemotherapy were considered.

Matted nodes exhibited a statistical tendency in their association with survival (47.7% vs. 60%, *p* = 0.096). Multivariate analysis discerned that independent factors contributing to survival were clinical stage and there was use of adjuvant chemotherapy (Table 3).

## 4. Discussion

In our series of 255 cases of CC with lymph node metastasis, we found that matted nodes presented in 18.4%, and their presence was associated with LVI and with a higher pN stage. Five-year survival of patients with matted nodes was 47.7%, compared with 60% in patient without matted nodes (0.096); however, this has only statistical tendency, and the multivariable analysis demonstrated that they are not associated with the survival. 

Colon cancer stands as a significant global health concern, with recent data from Globocan indicating its prevalence, constituting 11.9% of cases in Mexico and impacting both men and women at rates of 6.1% [1]. Within CC, the presence of lymph node metastases serves as a well-established prognostic indicator, offering crucial insights into disease progression. However, recent attention has shifted towards a specific aspect of lymph node involvement—matted lymph nodes—and their potential impact on patient outcomes. Extracapsular lymph node extension plays a central role in modifying CC prognosis [9]. This extracapsular involvement has been associated with younger age, advanced tumor stage, LVI, and PNI. Notably, matted lymph nodes, defined as clusters of fused nodes adhered by neoplastic cells, represent a distinctive form of ENE. While ENE has been explored across various malignancies, limited evidence is available regarding the prognostic implications of matted lymph nodes in CC [10].

Matted nodes have not been well studied in colon cancer, possibly due to the lack of clear and precise definitions. In many studies, they have been considered together with ENE of the tumor from a lymph node metastasis of adenocarcinoma, and the terms have been used interchangeably in the literature. ENE refers to the progression of cancer from within a lymph node (LN) outward into surrounding perinodal tissues. The pivotal event involves the disruption and infiltration of the tumor through the complete thickness of the LN capsule, which ordinarily acts as a barrier preventing tumor spread. This process is fundamental to the diagnosis and classification of ENE. The concept of ENE was initially documented in 1930 through a retrospective analysis of autopsy specimens from 20 patients with head and neck cancer [11]. Nearly thirty years later, the adverse prognostic implications of ENE were demonstrated in breast cancer, with similar findings later confirmed in head and neck squamous cell carcinoma (HNSCC) [12].

The underlying pathophysiology of ENE and matted nodes remains incompletely understood. Nevertheless, its presence frequently correlates with an aggressive cancer phenotype in other malignancies [13,14,15]. 

Our study contributes to the evolving understanding of lymph node involvement by investigating matted lymph nodes specifically. In other cancer types, matted nodes have been identified as an independent prognostic factor associated with adverse outcomes. For instance, in a systematic review, it was found that patients with matted nodes face up to a 1.6-fold higher risk of mortality compared to those without [4]. Addressing a gap in the existing literature, this study sought to elucidate whether the presence of matted lymph nodes independently correlates with reduced survival in patients diagnosed with CC. By focusing on this distinctive form of lymph node involvement, this study aimed to provide valuable insights into its prognostic implications and potential influence on treatment strategies. Our findings reveal that patients with matted lymph nodes are associated with a higher pN stage and LVI. Furthermore, the analysis of factors linked to survival among patients with lymph node metastasis identified matted nodes as showing a statistical tendency in association with reduced survival. Multivariate analysis emphasized that clinical stage and the utilization of adjuvant chemotherapy were independent factors contributing to survival.

Our study has several limitations. As the study population is retrospective, it may be subject to inherent biases in the collection of historical data. Additionally, the exclusion of cases with initial distant metastases could bias the results towards a population with a better prognosis. The definition of matted nodes involves variability in histopathological interpretation among different pathologists, which could affect the consistency of the classification. The statistical significance of the association between matted nodes and survival is not strong (*p* = 0.096), suggesting that the results may be inconclusive. The study was conducted at a single institution, which could limit the generalizability of the results to other populations or clinical settings. Therefore, the interpretation of the results should be cautious due to the inherent limitations of the retrospective design and the potential uncontrolled confounding variables.

## 5. Conclusions

In conclusion, this study sheds light on the underexplored territory of matted lymph nodes in CC with lymph node metastasis, indicating their potential as significant prognostic indicators. The association of matted nodes with higher pN stage and reduced survival underscores the importance of considering this specific lymph node characteristic in treatment planning and prognostic assessments for CC patients. Our results did not search statistical significance, but a tendency was detected; therefore, precise further research and validation studies are warranted to solidify these findings and integrate matted lymph nodes into the broader context of CC management. 

## Figures and Tables

**Figure 1 medicina-60-01194-f001:**
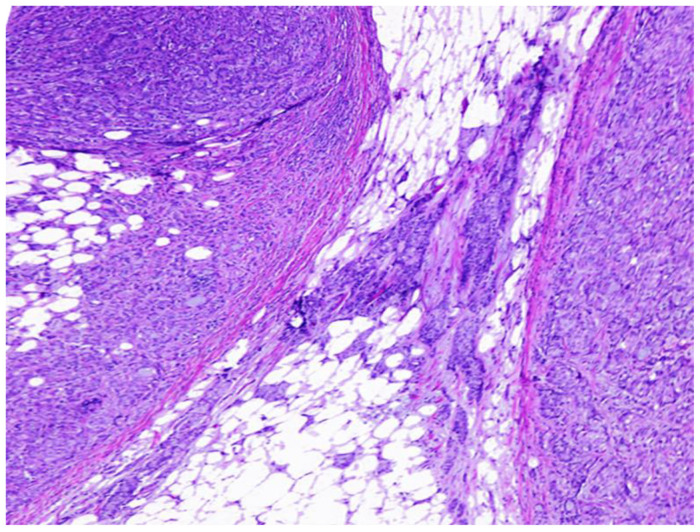
Hematoxylin and eosin-stained slide showing the presence of two contiguous nodes blended by neoplastic cells, which is the definition of matted nodes. Magnification: 40×.

**Table 1 medicina-60-01194-t001:** Clinicopathological characteristics of 255 cases of colon cancer with lymph node metastasis according to the presence of matted nodes.

Variable	No-Matted Nodes n = 208	Matted Nodes n = 47	*p*-Value *
Sex, n (%)			
Female	138 (52.7)	61 (50)	0.626
Male	124 (47.3)	61 (50)	
Age (years)–Median (IQR)	58 (48–68)	58 (48–69)	0.695
Location—N (%)			
Right	132 (63.5)	37 (78.7)	0.080
Left	76 (36.5)	10 (21.3)	
Tumoral diameter (mm), Median (IQR)	57 (40–80)	60 (50–80)	0.186
Resected lymph nodes, Median (IQR)	24 (18–32)	23 (18–32)	0.959
Metastatic lymph nodes, Median (IQR)	3 (1–6)	6 (4–8)	<0.001
Histologic grade, n (%)			
1	19 (9.1)	2 (4.3)	0.493
2	98 (47.1)	25 (53.2)	
3	91 (43.8)	20 (42.5)	
Lymph node status, n (%)			
pN1	118 (56.7)	4 (8.5)	<0.001
pN2	90 (43.3)	43 (91.5)	
Clinical stage, n (%)			
Stage III	137 (65.9)	28 (59.6)	0.415
Stage IV	71 (34.1)	19 (40.4)	
Lymphovascular invasion. n (%)			
No	85 (40.9)	11 (23.4)	0.026
Yes	123 (59.1)	36 (76.6)	
Venous invasion, n (%)			
No	132 (63.5)	33 (70.2)	0.382
Yes	76 (36.5)	14 (29.8)	
Perineural invasion, n (%)			
No	126 (60.6)	28 (59.6)	
Yes	82 (39.4)	19 (40.4)	0.899
Surgical margins, n (%)			
Negative	198 (95.2)	46 (97.9)	
Positive	10 (4.8)	1 (2.1)	0.414
Outcome, n (%)			
Alive	140 (67.3)	25 (53.2)	0.097
Dead	68 (32.7)	22 (46.8)	
Adjuvance, N (%)			
No	34 (16.3)	7 (14.9)	0.807
Yes	174 (83.7)	40 (85.1)	
Subtype			
Not otherwise specified	94 (45.2)	24 (51.1)	0.823
Mucinous	37 (17.8)	6 (12.8)	
Other	77 (5.8)	17 (36.2)	

* Chi square test for categorical variables. Student’s *t*-test for numerical variables; IQR = interquartile range.

**Table 2 medicina-60-01194-t002:** Clinicopathological characteristics associated with survival of 255 cases of colon cancer with lymph node metastasis.

Variable	5-Year Overall Survival (%)	*p*-Value *
Sex		
Female	53.8	0.066
Male	61.7	
Location		
Right	57.9	0.556
Left	57.7	
Histologic grade		
1	78.6	0.162
2	55.9	
3	54.9	
Lymph node status		
pN1	69.9	<0.001
pN2	45.7	
Clinical stage		
Stage III	71.0	<0.001
Stage IV	28.1	
Lymphovascular invasion.		
No	69.4	0.004
Yes	50.4	
Venous invasion		
No	63.6	0.034
Yes	45.7	
Perineural invasion		
No	65.8	0.005
Yes	44.9	
Surgical margins		
Negative	58.8	0.242
Positive	34.3	
Adjuvance		
No	37.9	<0.001
Yes	61.5	
Matted nodes		
No	60	0.096
Yes	47.7	

* Log-rank test.

**Table 3 medicina-60-01194-t003:** Multivariate analysis of factors associated with survival in 255 cases of colon cancer with lymph node metastasis.

Variable	Chi Square Value	Hazard Ratio (95% Confidence Intervale)	*p*-Value
Clinical stage (III vs. IV)	24.509	3.104 (1.982–4.859)	<0.001
Adjuvance (No vs. Yes)	23.422	0.272 (0.161–0.461)	<0.001
Lymph node status (pN1 vs. pN2)	3.582	1.602 (0.983–2.610)	0.058
Lymphovascular invasion (No vs. Yes)	3.157	1.669 (0.949–2.936)	0.076
Matted nodes (No vs. Yes)	0.805	1.278 (0.748–2.186)	0.369
Perineural invasion (No vs. Yes)	0.387	1.162 (0.724–1.864)	0.534
Venous invasion (No vs. Yes)	0.050	0.945 (0.575–1.553)	0.823

## Data Availability

Data of the study are available freely by reasonable request via email.

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
