# Peer review of "Does the Presence of Matted Nodes in Colon Adenocarcinoma Influence 5-Year Overall Survival?"

_medicina, 2024, doi:10.3390/medicina60081194_

Round 1

Reviewer 1 Report

Comments and Suggestions for Authors

Authors presented the sudy on "Does The Presence of Matted Nodes In Colon Adenocarcinoma 2 Influence 5-Year Overall Survival?"

1. Authors havent included much background information Introduction section. for example categerized the study into   lymph node metastasis alone , and 2) lymph node metastasis with matted nodes. No information was provided on matted nodes or why they choose to do these two categeries.

2. why the authoros choose 255 cases with lymph node metastsis and only 47 for matted node categerie. 

3.what will be the status of the patients who doesnt have colectomy. if you can add the information into the table 1 it will realy appropriate to compare with other two categeries. 

4. CRC was abbrevated in the introduction ( Line 35), instead of writing Colorectal cancer authors can use CRC in line 138. same with other abbrevations please check through the paper. 

5. in the conclusion section also authors mentioned " study sheds light on the underexplored territory of matted lymph 167 nodes in colon cancer" with out adding the data from other group that doesnt have lymph node metastasis you cant use statements like this. 

Author Response

Thank you very much for your comments, which will undoubtedly improve our manuscript.

Comment 1. Authors havent included much background information Introduction section. for example categerized the study into   lymph node metastasis alone , and 2) lymph node metastasis with matted nodes. No information was provided on matted nodes or why they choose to do these two categeries.

Reply 1. We understand that the introduction is somewhat brief, but the information regarding the study of matted nodes in colon cancer is sparse in the literature. This scarcity of specific studies is why there is limited information available; matted nodes have not been studied independently. Therefore, our study focuses solely on comparing these groups. Among patients with nodal metastasis, we divide them into two distinct groups: those with matted nodes and those without. We have elaborated further on this aspect in the introduction and rephrased it for clarity.

Comment 2. why the authoros choose 255 cases with lymph node metastsis and only 47 for matted node categerie. 

Reply 2. We selected these cases because, from the total number of consecutively treated colectomy cases, we only needed to study those with lymph node metastases. This was due to our aim of determining whether the presence of matted nodes in these lymph nodes is an adverse prognostic factor compared to the mere presence of lymph node metastases. We believe this is clear in the methodology; however, we have reviewed and rewritten parts of the abstract and methodology for greater clarity.

Comment 3. what will be the status of the patients who doesnt have colectomy. if you can add the information into the table 1 it will realy appropriate to compare with other two categeries. 

Reply 4. All cases have colectomy, our study group is patients that received colectomy for colon cancer and presented lymph node metastasis identified in the pathology analysis. We amend the methodology, and we re-write for clarity.

Comment 4. CRC was abbrevated in the introduction ( Line 35), instead of writing Colorectal cancer authors can use CRC in line 138. same with other abbrevations please check through the paper. 

Reply 4: We agree and amend this. 

Comment 5. In the conclusion section also authors mentioned " study sheds light on the underexplored territory of matted lymph 167 nodes in colon cancer" with out adding the data from other group that doesnt have lymph node metastasis you cant use statements like this. 

Reply 5. We agree and amend this. 

Reviewer 2 Report

Comments and Suggestions for Authors

In the present manuscript the authors analize the prognostic significance on survival of matted lymph nodes in patients with colon cancer. The manuscript is interesting as the prognostic value of this histophatologic factors has been scarcely evaluated in colon cancers. 

Some modifications may improve the manuscript:

1- Abstract. Results (line 24). "The 5-year survival rate for patients with matted nodes was 47,7%, compared to 60% without, however, this association demonstrated only a statistical tendency").

The authors must provide statistical data on the significance of the comparison.

2. Abstract. Conclusion (line 27-29). "This study underscores matted nodes as potential prognostic indicators in colon cancer.."

The authors should indicate that althoug the patients with matted nodes showed lower survival, this figure did not search statistical significance, but a tendency was detected, which precise further research.

3. Material and methods

The authors should indicate the design of the study: retrospective and observational.

4. Material and Methods. Was this study aproved by any Institutional Review Board?

Results. Line 122. Comparison of 5-year survival of patients with matted nodes vs patients without matted nodes (and data of statistical significance) must be provided  

Comments on the Quality of English Language

English language need only minor revision

Author Response

Comment 1- Abstract. Results (line 24). "The 5-year survival rate for patients with matted nodes was 47,7%, compared to 60% without, however, this association demonstrated only a statistical tendency").

The authors must provide statistical data on the significance of the comparison.

Reply 1: we agree and amend it.

Comment 2. Abstract. Conclusion (line 27-29). "This study underscores matted nodes as potential prognostic indicators in colon cancer.."

The authors should indicate that althoug the patients with matted nodes showed lower survival, this figure did not search statistical significance, but a tendency was detected, which precise further research.

Reply 2. We agree and amend this in this section (abstract) and also in the conclusion. 

Comment 3. Material and methods

The authors should indicate the design of the study: retrospective and observational.

Reply 3. We amend this.

Comment 4. Material and Methods. Was this study aproved by any Institutional Review Board?

Reply 4. Yes, and we added this information in the methods section.

Comment 5. Results. Line 122. Comparison of 5-year survival of patients with matted nodes vs patients without matted nodes (and data of statistical significance) must be provided  

Reply 4. We amend this.

Round 2

Reviewer 1 Report

Comments and Suggestions for Authors

I do not have any further comments 

Author Response

Nothing to reply.

Reviewer 2 Report

Comments and Suggestions for Authors

The manuscript has been sufficiently improved to warrant publication

Comments on the Quality of English Language

The english languaje is correct and only needs minor corrections

Author Response

Nothing to reply.